# Effect of Rowachol on the Gallbladder Dysmotility Disorder Based on Gallbladder Ejection Fraction

**DOI:** 10.3390/medicina59010105

**Published:** 2023-01-03

**Authors:** Sang Hoon Lee, Hyun Woo Chung, Tae Yoon Lee, Young Koog Cheon

**Affiliations:** 1Department of Internal Medicine, Konkuk University Medical Center, Konkuk University School of Medicine, Seoul 05030, Republic of Korea; 2Department of Nuclear Medicine, Konkuk University Medical Center, Konkuk University School of Medicine, Seoul 05030, Republic of Korea

**Keywords:** terpenes, biliary dyskinesia, gallbladder disease

## Abstract

*Background and Objectives*: Although laparoscopic cholecystectomy is the preferred treatment method in patients who experience typical biliary pain with or without gallstones, medical treatment has not been extensively studied. Rowachol is a potent choleretic agent, comprising six cyclic monoterpenes. This study aimed to investigate the clinical improvement and changes in gallbladder ejection fraction (GBEF) by Rowachol treatment in patients with typical biliary pain. *Materials and Methods*: We retrospectively reviewed 138 patients with typical biliary pain who underwent cholescintigraphy from July 2016 to April 2022. We included patients who received Rowachol for more than 2 months and underwent follow-up GBEF measurements. Finally, we analyzed pre- and post-treatment symptoms and GBEF. GBEF was calculated using the fatty meal-stimulated cholescintigraphy. *Results*: This retrospective observational study included 31 patients; their median age was 46.0 (range, 26.0–72.7) years, and 22 (71.0%) were female. Overall, 9 (29.0%) patients had gallbladder stones or sludges (maximum size: 2 mm) on initial transabdominal ultrasonography. During a median follow-up of 23.3 months, the symptoms of 21 (67.7%) patients were resolved after a median Rowachol treatment of 10.0 months. The mean GBEF was significantly improved after Rowachol treatment (initial cholescintigraphy: 42.6% ± 16.2%; follow-up cholescintigraphy: 53.0% ± 18.1%, *p* = 0.012). In patients with a GBEF ≤35% (n = 9), Rowachol significantly increased the GBEF from 21.3% ± 8.3% to 49.1% ± 20.7% (*p* = 0.008). *Conclusions*: Rowachol may have beneficial medical effects that can improve gallbladder dysfunction and treatment response.

## 1. Introduction

Typical biliary-type abdominal pain is a commonly encountered clinical problem, which is defined as moderate to severe steady pain, located in the epigastric and/or right upper quadrant, lasting for >30 minutes, according to the Rome IV criteria [1]. Large epidemiologic studies have estimated the prevalence of biliary-type pain with or without gallstones to be approximately 3.7–7.7% [2,3]. Biliary pain is sometimes hardly distinguished from pain occurring in high-prevalence conditions, such as irritable bowel syndrome, gastroesophageal reflux disease, functional dyspepsia, and cholelithiasis in the gallbladder (GB) and biliary tract [4].

Functional GB disorder is a relatively rare condition of biliary pain, and is defined as biliary pain in the absence of gallstones or other structural diseases. The diagnosis of functional GB disorder is supported by a low GB ejection fraction (GBEF) on cholescintigraphy, and normal laboratory test results. Furthermore, a low GBEF, especially <35%, is suggested as a clinical predictor for symptomatic relief after cholecystectomy in patients with functional GB disorder [5,6].

GB dysfunction is involved in the pathogenesis of patients with biliary pain, including functional GB disorder, microlithiasis, and symptomatic gallstone disease [1,7]. Impaired GB emptying, as reflected by a decreased GBEF, promotes bile stasis and inflammation in the GB, and these vicious cycles of GB dysfunction may eventually lead to gallstone formation and symptoms over a period of time [1,7]. Although the symptoms of the disease spectrum of GB dyskinesia appear to be indistinguishable from one another, cholecystectomy is the preferred treatment option for both symptomatic gallstones and functional GB disorder. However, receiving surgery as the primary treatment is burdensome for patients, especially the elderly and those with comorbidities, and for these patients, medical treatment is alternatively considered. To date, medical treatment has not been extensively studied for these typical biliary pain patients with or without gallstones.

Rowachol is a potent choleretic agent, consisting of six cyclic monoterpenes, all of which are derived from purified plant essential oils. Each terpene in Rowachol has not only choleretic effects, but also anti-bacterial, anti-oxidant, anti-inflammatory, and anti-spasmodic activity in vitro and in vivo [8,9]. Moreover, Rowachol has shown significant symptom relief in patients with gallstones [10,11,12]. We hypothesized that Rowachol treatment for patients with biliary pain improves their symptoms and GB dysfunction, which may be directly measured by changes in the GBEF.

This study aimed to investigate the clinical improvement and changes in GBEF caused by Rowachol treatment in patients with typical biliary pain.

## 2. Materials and Methods

### 2.1. Study Population

From July 2016 to April 2022, 138 patients were retrospectively included in this single-center pilot study, performed at the Konkuk University Medical Center (Korea). The inclusion criteria were as follows: (1) initial complaint of typical biliary pain according to the Rome IV criteria; (2) Rowachol prescription for >2 months; and (3) initial and follow-up measurements of the GBEF, performed by cholescintigraphy (Figure 1). For subgroup analysis, the included patients were divided into two groups, based on a cutoff GBEF value of 35% on initial cholescintigraphy. All the clinical, laboratory, and radiologic data were collected from electronic medical records and were reviewed retrospectively. The STROBE guidelines were used to ensure the reporting of this observational study [13].

### 2.2. Treatment and Clinical Assessment

Rowachol (Rowa Pharma, Cork, UK), as a mixture of terpene (pinene 17 mg, camphene 5 mg, cineol 2 mg, menthone 6 mg, menthol 32 mg, and borneol 5 mg), was prescribed orally at a dose of 100 mg, three times a day, for at least 2 months. The follow-up interval and post-treatment GBEF measurements were dependent on the physician. All patients’ pain scales were assessed at each outpatient visit, according to the visual analog scale. Clinical response was evaluated after medical treatment as follows: “no response” indicates worsening, no change, or continuing symptoms; “partial response” indicates slight, moderate, better, or <75% improved symptom; and “resolved” indicates asymptomatic, cured, or >75% improved status [14].

### 2.3. Cholescintigraphy

All the patients fasted for at least 4 hours before the intravenous administration of approximately 185 MBq of 99mTc-mebrofenin (bromo-2,4,6-trimethylacetanilido iminodiacetic acid). Cholescintigraphy was performed using a large-field-of-view dual-head γ-camera, equipped with a low-energy high-resolution collimator (E.CAM, Siemens Medical Solutions, Issaquah, WA, USA), for 60 min, with the patient in the supine position. With visualization of the GB filling, the patients ingested a fatty meal (200 mL of milk and two slices of cheese) while sitting. Patients lay in the supine position again, and images were obtained for 60 min. For the GBEF, regions of interest were drawn for the GB and adjacent liver (background) using the Syngo workstation software (version 2008, Siemens, Germany). Background- and decay-corrected GB counts were generated. GBEF was calculated as follows: (maximum GB counts − minimum GB counts)/maximum GB counts × 100.

### 2.4. Statistical Analysis

Data are presented as mean ± standard deviation, median (range), or *n* (%), as appropriate. The paired *t*-test was used for comparing the changes in the GBEFs at the initial and follow-up (i.e., after Rowachol treatment) measurements. A two-tailed *p*-value of 0.05 was considered statistically significant, and the statistical analysis was performed using the SPSS version 17.0 (PASW Statistics Inc., Chicago, IL, USA).

## 3. Results

### 3.1. Baseline Characteristics

A total of 31 patients were included in the final analysis (Figure 1). The baseline characteristics of the 31 patients are summarized in Table 1. The median age of patients was 46.0 (range, 26.0–72.7) years, and females were predominant (71.0%). All the patients were examined by transabdominal ultrasonography, and 9 (29.0%) patients had GB stones or sludges. The ultrasonographic findings of these patients were as follows: 2 mm-sized single gallstone (*n* = 1), multiple 1~2-mm-sized gallstones (*n* = 4), and sandy GB sludge (*n* = 4). The baseline laboratory findings, including liver function test, were not remarkable for all the patients. In addition, 28 (90.1%) patients reported that the recent endoscopic findings were not significant enough to cause symptoms. The co-medications during the Rowachol treatment period are described in Appendix A.

### 3.2. Clinical Outcomes

After a median Rowachol treatment of 10.0 months, the symptoms of 21 (67.7%) patients were resolved (Table 2). Among them, three patients finally underwent cholecystectomy, due to symptom recurrence after stopping Rowachol treatment (*n* = 2), and aggravation of GB wall thickening (*n* = 1). Among five (16.1%) patients with no response, four patients finally received cholecystectomy (Figure 2). There was no patient who received the endoscopic sphincterotomy for sphincter of Oddi disease. Additionally, no one discontinued Rowachol due to the side effects.

### 3.3. Results of Cholescintigraphy

On initial cholescintigraphy, the mean baseline GBEF was 45.8% (range, 6.5–68.2%), and 9 (29.0%) patients had a GBEF of ≤35% (Table 1). During a median follow-up of 23.3 months, the timings of follow-up GBEF measurements were variable from the end date of Rowachol treatment (Figure 2). The median time interval between the initial and follow-up cholescintigraphy was 3.5 (range, 2.6–32.6) months.

Changes in GBEF were demonstrated as boxplots in Figure 3. The mean GBEFs at the initial and follow-up measurements was 42.6% ± 16.2% and 53.0% ± 18.1%, respectively (Table 3). The 10.4% change in GBEF was significant (*p* = 0.012). Figure 4 shows representative cholescintigraphy of patient with improved GBEF (21.1% to 61.4%).

### 3.4. Subgroups Analysis Based on Initial Cholescintigraphy and Clinical Outcomes

In patients with a GBEF ≤35% (*n* = 9), Rowachol significantly increased the GBEF from 21.3% ± 8.3% to 49.1% ± 20.7% (*p* = 0.008, Table 3). However, the rate of improvement in the symptoms after Rowachol treatment did not differ significantly between the patients with a GBEF ≤35% and patients with a GBEF > 35% (77.8% vs. 63.6%, *p* = 0.445).

Table 4 summarized the comparison among subgroups, according to clinical outcomes. Except for the proportion of cholecystectomy, there was no significant difference among them.

## 4. Discussion

Rowachol may have beneficial medical effects, in terms of improving the symptoms and GBEF in this retrospective pilot study. Rowachol resolved the symptoms of 21 of the 31 patients (67.7%) with typical biliary pain, and significantly increased their GBEF (10.4%, *p* = 0.012). This finding provides the first evidence in favor of the hypothesis that Rowachol is a potential complementary treatment for gallbladder dysfunction.

Rowachol, a mixture of terpene, was first introduced as a cholelitholytic agent for gallstone dissolution therapy [15,16]. It inhibits hepatic hydroxymethyl glutamylcoenzyme A reductase, alters biliary cholesterol saturation, and lowers the lithogenic index of human bile [8,17,18]. Menthol, one of the major monoterpenes of Rowachol, is mainly secreted into bile in the form of menthol glucuronide, which is known to enhance the solubility of both calcium carbonate and phosphate, thus helping to reduce the calcified concretions of gallstone [19]. Furthermore, each component of Rowachol has mechanisms of anti-oxidant, anti-septic, anti-spastic, and analgesic effects in vitro [9]. Several clinical studies have reported clinical improvement with long-term Rowachol treatment in patients with gallstones [10,11,12]. However, to the best of our knowledge, this study is the first to investigate the therapeutic effects of Rowachol in a study group of patients with typical biliary pain, including functional GB disorder.

Recently, other litholytic agents, such as ursodeoxycholic acid and chenodeoxycholic acid, showed improvements in symptoms and GBEF in most patients with functional GB disorder [20]. Kim et al. suggested that litholytic agents improve the patients’ GBEF and symptoms by dissolution of microlithiasis. In fact, the majority of patients (77.4%) in our study also concurrently received these bile acid preparations (Appendix A). Theoretically, Rowachol and each bile acid have different therapeutic mechanisms and a synergistic effect on gallstone dissolution [21]. In further investigation for follow-up ultrasound findings in the study group, among the nine patients who initially had GB stones or sludges, three patients (two GB sludges; one 2 mm-sized gallstone) resolved the disease after Rowachol administration. A prospective large-scale study with a control group is needed to reveal the actual effects of Rowachol as a medical treatment option.

GB dysfunction can contribute to promoting GB inflammation and stasis and eventually triggering patients’ symptoms [1]. In a previous study, GBEF was significantly higher in the following order: healthy volunteers, microlithiasis, and the gallstone patient group [7]. The spectrum of biliary diseases has been described as leading to crystal growth and subsequent gallstone formation and chronic inflammation in GB, due to bile saturation and GB dysfunction [22]. Biliary pain can probably occur at any point in this disease spectrum. In our study group, 9 (29.0%) patients were accompanied by GB stones or sludges of up to 2 mm in size. There was no significant difference in baseline characteristics, clinical outcomes, or GBEF changes between patients with gallstones or sludges and the rest of the patients.

This study has several limitations. First, the nature of the retrospective pilot study conducted at a single center without any control group made it difficult to exclude the placebo effect, selection bias, or other confounding factors. In addition, most patients were excluded from the study because the examination and treatment strategies were not unified, so they did not receive Rowachol treatment or had no follow-up GBEF measurements. Even within the study group, treatment duration and timing of follow-up cholescintigraphy were widely distributed, which can confuse the research results. Furthermore, because of the small sample size, it is difficult to confirm a definitive conclusion about the medical effect of Rowachol on patients with these disease spectra. Prospective research with established protocols is warranted. As mentioned earlier, many patients were prescribed medications other than Rowachol; therefore, we cannot determine whether this medical treatment is superior to the others. Lastly, our study group may be heterogeneous in the cause of biliary dysfunction, given that other conditions, such as sphincter of Oddi dysfunction and duodenal hypersensitivity, can cause similar biliary pain [23].

## 5. Conclusions

In conclusion, this study suggests that Rowachol can relieve symptoms and modulate GBEF in patients with biliary pain, especially in those with a reduced GBEF. In the future, we hope that Rowachol serves as a potential complementary treatment option for patients with GB dysfunction.

## Figures and Tables

**Figure 1 medicina-59-00105-f001:**
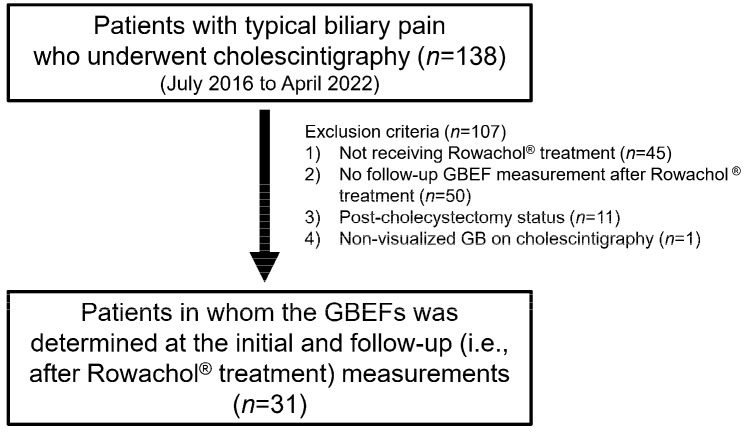
Flow chart for the study population. GBEF, gallbladder ejection fraction; GB, gallbladder.

**Figure 2 medicina-59-00105-f002:**
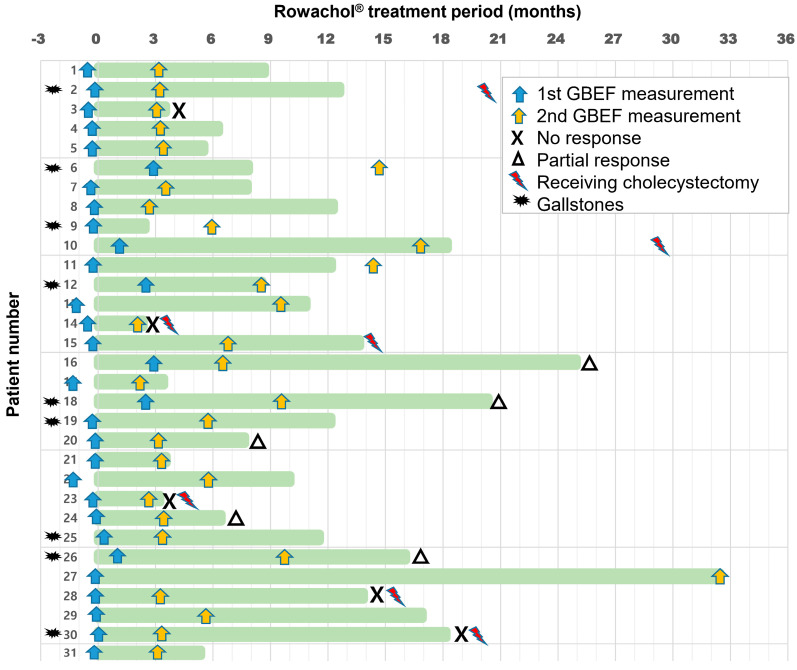
Bar chart for the timetable of GBEF measurement and Rowachol treatment. The horizontal axis on the top indicates time interval (months) from the start date of Rowachol treatment. Rowachol treatment period is defined as the length of green bar on each patient of the vertical axis. Blue and yellow arrows indicate the time points of initial and follow-up GBEF measurements. For clinical response, “X” on the right end of the green bar indicates “no response” and “triangle” indicates “partial response” for each patient. Among five (16.1%) patients with no response, four finally received cholecystectomy (red lightning bolt). The other three patients finally underwent cholecystectomy, due to symptom recurrence after stopping Rowachol treatment (Case 2 and 10), and aggravation of GB wall thickening (Case 15). Additionally, the black explosive star indicates the nine patients with GB stones or sludge on initial transabdominal ultrasonography examination. GBEF, gallbladder ejection fraction; GB, gallbladder.

**Figure 3 medicina-59-00105-f003:**
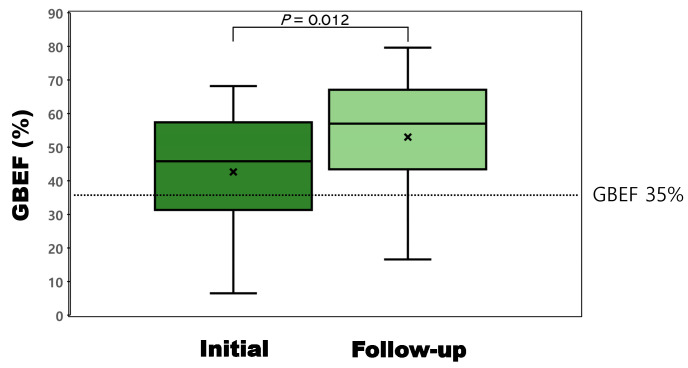
Boxplots for the GBEFs at the initial and follow-up measurements. The mean GBEFs at the initial and follow-up measurements were 42.6% (range, 6.5–68.2%) and 53.0% (range, 16.6–79.6%), respectively. The 10.4% change in GBEF was significant (*p* = 0.012). GBEF, gallbladder ejection fraction.

**Figure 4 medicina-59-00105-f004:**
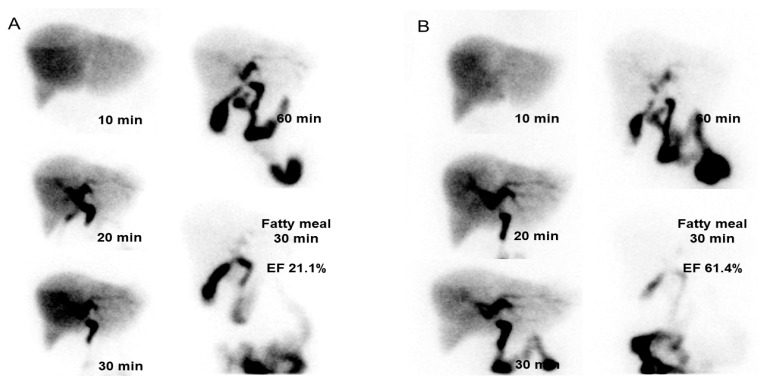
Representative cholescintigraphy of a patient with improved GBEF. (**A**) On initial cholescintigraphy, and (**B**) follow-up cholescintigraphy, GBEF was measured as 21.1% and 61.4%, respectively. GBEF, gallbladder ejection fraction.

**Table 1 medicina-59-00105-t001:** Baseline characteristics (*n* = 31).

Characteristics	
Age, year	46.0 (26.0–72.7)
Gender, female	22 (71.0%)
Laboratory results	
AST, IU/L	26.5 ± 9.3
ALT, IU/L	21.5 ± 11.7
Total bilirubin, mg/dL	0.64 ± 0.27
Transabdominal ultrasonography	
GB stones and/or sludges ^1^	9 (29.0%)
Scintigraphy	
GBEF, %	45.8 (6.5–68.2)
Patients with a GBEF ≤35%	9 (29.0%)
Follow-up duration, months	23.3 (4.1–50.2)

^1^ Maximum size of gallstones: 2 mm; Values shown are *n* (%) or median (range). AST, aspartate aminotransferase; ALT, alanine aminotransferase; GB, gallbladder; GBEF. gallbladder ejection fraction.

**Table 2 medicina-59-00105-t002:** Clinical outcomes (*n* = 31).

Variables	
Treatment duration, months	10.0 (2.4–32.5)
Clinical response	
No response	5 (16.1%)
Partial response	5 (16.1%)
Resolved	21 (67.7%)
Underwent cholecystectomy ^1^	7 (22.6%)

^1^ Including patients with no response (*n* = 4), symptom recurrence after stopping Rowachol treatment (*n* = 2), and aggravation of gallbladder wall thickening (*n* = 1). Values shown are *n* (%) or median (range).

**Table 3 medicina-59-00105-t003:** Comparison of the GBEFs at the initial and follow-up (after Rowachol treatment) measurements.

	Initial (%)	Follow-Up (%)	*p*-Value
All patients (*n* = 31)	42.6 ± 16.2	53.0 ± 18.1	0.012
Patients with an initial GBEF ≤35% (*n* = 9)	21.3 ± 8.3	49.1 ± 20.7	0.008
Patients with an initial GBEF >35% (*n* = 22)	51.4 ± 8.5	54.6 ± 17	0.367

Values shown are means ± SD. GBEF, gallbladder ejection fraction.

**Table 4 medicina-59-00105-t004:** Comparison of patients according to clinical outcomes.

Characteristics	Resolved (*n* = 21)	Partial Response (*n* = 5)	No Response (*n* = 5)	*p*-Value
Age, year	45.4 (26.0–72.7)	46.0 (28.0–53.9)	55.3 (43.0–63.5)	0.359
Gender, female	13 (61.9%)	4 (80.0%)	5 (100%)	0.214
Transabdominal ultrasonography				
GB stones and/or sludges ^1^	6 (28.6%)	2 (40.0%)	1 (20.0%)	0.782
Scintigraphy				
Initial GBEF, %	41.6 ± 17.1	52.3 ± 8.2	37.3 ± 16.6	0.240
Follow-up GBEF, %	57.2 ± 16.4	50.8 ± 20.1	37.5 ± 17.5	0.109
GBEF change, %	15.6 ± 20.4	−1.5 ± 16.8	0.2 ± 26.3	0.112
Treatment duration, months	10.0 (2.5–32.5)	16.1 (6.4–25.0)	3.5 (2.4–18.2)	0.257
Underwent cholecystectomy	2 (9.5%)	0 (0.0%)	4 (80.0%)	0.001
Follow-up duration, months	23.4 (4.3–50.2)	23.3 (4.1–40.1)	5.9 (4.2–29.4)	0.435

Values shown are *n* (%) or median (range). AST, aspartate aminotransferase; ALT, alanine aminotransferase; GB, gallbladder; GBEF, gallbladder ejection fraction.

## Data Availability

Not applicable.

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
