# Peer review of "Effect of Rowachol on the Gallbladder Dysmotility Disorder Based on Gallbladder Ejection Fraction"

_medicina, 2023, doi:10.3390/medicina59010105_

Round 1

Reviewer 1 Report

The authors presented interesting results that suggest Rowachol may be beneficial for improving biliary pain from GB dysfunction. Although this study has limitation of retrospective nature and small sample size, it could provide clinically relevant clue and warrant future prospective trial.

I would raise some questions and comments.

#1. To classify treatment response, the authors provided quantitative criteria in change of symptom such as 75% improvement. How did you quantitatively measure the patient`s symptom?

#2. How many patients underwent ultrasonography after Rowachol treatment? Is there any change of size or presence of gallbladder stone after treatment?

#3. According to the figure 2, most of patients who were in “Resolved” group took Rowachol no more than 6~9 months, while those in “Partial response” group took Rowachol more than 15 months. What is the authors opinion on when is the optimal period to evaluate treatment response to decide to continue or stop the treatment?

#4. Was there any difference in GBEF change between clinical response groups? Was the significant change of GBEF related to symptom relief?

#5. In page 6 line 180, the authors stated their study was underwent in patient with “functional GB disorder”. However, according to the definition, which the authors stated in page 1 line 39, functional GB disorder is defined as biliary pain in the absence of gallstones or other structural diseases. Some patients in this study had gallbladder stone, so this should be corrected.

Author Response

Dec 26, 2022

We would like to thank you for taking your time and giving valuable comments on our submission.

We adopted the reviewers` comments and revised the manuscript accordingly. Please refer to the answers below for changes of the revised manuscript. Also, all changes in the revised manuscript are marked up using the “Track Changes” function.

Thank you again for kindly reviewing our manuscript.

Sincerely,

Reviewer 1.

The authors presented interesting results that suggest Rowachol may be beneficial for improving biliary pain from GB dysfunction. Although this study has limitation of retrospective nature and small sample size, it could provide clinically relevant clue and warrant future prospective trial.

I would raise some questions and comments.

#1. To classify treatment response, the authors provided quantitative criteria in change of symptom such as 75% improvement. How did you quantitatively measure the patient`s symptom?

Answer: Thank you for your valuable comment. The patient’s biliary pain intensity was measured according to the visual analogue scale (VAS). Patients’ VAS scores were assessed at each outpatient visit. We inserted the following sentence in the “Materials and methods – Treatment and clinical assessment” section in the revised manuscript: “All patients’ pain scale were assessed at each outpatient visit according to the visual analogue scale.” (Page 2, line 79-81)

#2. How many patients underwent ultrasonography after Rowachol treatment? Is there any change of size or presence of gallbladder stone after treatment?

Answer: Thank you for your relevant question. Except for two patients who underwent cholecystectomy before follow-up ultrasonography after Rowachol treatment, 58.6% (17/29) of patients performed the ultrasonography. Among 9 patients who initially had GB stones or sludges, 3 (33.3%) patients (two patients for GB sludges; one patient for 2mm-sized gallstone) resolved their disease after Rowachol administration. The results of the other 6 patients have not changed. Therefore, we decided to insert the following sentence in “Discussion” section as follows; “In further investigation for follow-up ultrasound findings in the study group, among the nine patients who initially had GB stones or sludges, three patients (two GB sludges; one 2mm-sized gallstone) resolved the disease after Rowachol administration.” (Page 7, line 204~207). Thanks again for your comment.

#3. According to the figure 2, most of patients who were in “Resolved” group took Rowachol no more than 6~9 months, while those in “Partial response” group took Rowachol more than 15 months. What is the authors opinion on when is the optimal period to evaluate treatment response to decide to continue or stop the treatment?

Answer: Thank you for your meaningful and constructive comment. In many previous observational and prospective studies for patients with gallstone disease, including difficult-to-remove CBD stones, and post-cholecystectomy pain, the treatment duration of Rowachol ranged from at least 3 to 24 months. Most studies represented a treatment period of 6 to 12 months. In our personal opinions, Rowachol treatment is usually recommended for more than 6 months, and if patients’ responses were not appropriate after 12 months of treatment, additional examinations or other treatment options such as surgery have been considered. As you mentioned, patients with partial response were the most difficult to manage in real clinical situations, and we suggested that this clinical burden needs to be investigated through further studies in the future. In addition, we added this limitation as follows in “Discussion” section; “Even within the study group, treatment duration and timing of follow-up cholescintigraphy were widely distributed, which can confuse the research results” (Page 7, line 225-226).

#4. Was there any difference in GBEF change between clinical response groups? Was the significant change of GBEF related to symptom relief?

Answer: Thank you for your valuable comment. We could not find the statistical difference of GBEF change among subgroups based on clinical response, and we summarized the comparative results of patients according to clinical outcomes as Table 4 (Page 6, line 172-178). In addition, in the “resolved” group, GBEF improvement seemed to be higher than the other groups, but it was not statistically significant (p = 0.112) because of small numbers of each group.

#5. In page 6 line 180, the authors stated their study was underwent in patient with “functional GB disorder”. However, according to the definition, which the authors stated in page 1 line 39, functional GB disorder is defined as biliary pain in the absence of gallstones or other structural diseases. Some patients in this study had gallbladder stone, so this should be corrected.

Answer: Thank you for pointing this out. As you mentioned, we corrected this sentence as follows (highlighted in yellow); However, to the best of our knowledge, this study is the first to investigate the therapeutic effects of Rowachol in a study group of patients with typical biliary pain, including functional GB disorder”. (Page 7, line 196-197)

Reviewer 2 Report

This pilot study evaluated the efficacy of Rowachol for biliary pain (GB disorder). Because of the rack of evidence of medication treatment for biliary pain, this study is interesting and important for treatment. However, it is necessary to consider the following points.

Major

#1

In this study, patients with biliary pain were included.

Gastritis or esophagitis can cause epigastralgia pain which is similar to biliary pain.

It is necessary to describe how to exclude diseases other than GB disorders (biliary pain).

#2

Is there any case who received endoscopic sphincterotomy after or during Rowachole administration?

#3

Is there any case who administered NSAIDs or proton pump inhibitors after or during Rowachole administration?

If there are some cases in which symptoms have improved with NSAIDs, etc., it is necessary to examine the relationship with clinical outcomes.

#4

Although the biliary pain was improved after Rowachol administration, 3 cases (#2, #10, and #15 cases in Figure 2) were undergone cholecystectomy. Please explain the reason for these cases required surgical treatment.

#5

Please describe the adverse events after Rowachol administration.

#6

Because Rowachol has an anti-bacterial effect, it is necessary to consider the risk of drug-resistant bacteria. Please discuss the duration of administration of Rowachol and the timing of discontinuation.

#7

Six and 9 cases of Figure 2 were evaluated 2nd GBEF measurement after discontinuation of Rowachol at least 3 months.

It may be better to exclude these cases since they are thought to be evaluated after the effect of Rowachol has been lost.

Minor

#1

Please explain “Cholescitigrapy” using some figures.

Author Response

Dec 26, 2022

We would like to thank you for taking your time and giving valuable comments on our submission.

We adopted the reviewers` comments and revised the manuscript accordingly. Please refer to the answers below for changes of the revised manuscript. Also, all changes in the revised manuscript are marked up using the “Track Changes” function.

Thank you again for kindly reviewing our manuscript.

Sincerely,

Reviewer 2.

This pilot study evaluated the efficacy of Rowachol for biliary pain (GB disorder). Because of the rack of evidence of medication treatment for biliary pain, this study is interesting and important for treatment. However, it is necessary to consider the following points.

Major

#1. In this study, patients with biliary pain were included. Gastritis or esophagitis can cause epigastralgia pain which is similar to biliary pain. It is necessary to describe how to exclude diseases other than GB disorders (biliary pain).

Answer: Thank you for your important question. In real clinical practice, it is not easy to distinguish typical biliary pain from other causes of epigastric pain resulting from gastritis or esophagitis. Therefore, we have strictly applied the Rome IV criteria for biliary pain and checked the baseline esophagogastroscopy results. In fact, 28 (90.1%) patients who received upper endoscopy or had informed recent endoscopic findings, and it was confirmed that not all results were significant to cause symptoms. For preventing the concerns, you mentioned, we added the following sentence in “Results – Baseline characteristic” section as follows; “In addition, 28 (90.1%) patients were checked that not all the recent endoscopic findings were significant enough to cause symptoms.”. (Page 3, line 112-113)

#2. Is there any case who received endoscopic sphincterotomy after or during Rowachole administration?

Answer: Thank you for your meaningful comment. In our study group. there was no patient who received the endoscopic sphincterotomy for Sphincter of Oddi disease.

#3. Is there any case who administered NSAIDs or proton pump inhibitors after or during Rowachole administration? If there are some cases in which symptoms have improved with NSAIDs, etc., it is necessary to examine the relationship with clinical outcomes.

Answer: Thank you for your comments. The list of co-medications during the Rowachol treatment period was described in Supplementary Table 1, including 20 patients (64.5%) who took proton pump inhibitors. No one prescribed NSAIDs during Rowachol administration. We agree that these concomitant medications may confound our study’s outcome and described as a limitation of the study. Additionally, we couldn’t find any statistical difference in clinical outcomes, such as the proportion of “resolved” responses, when patients were divided according to whether they received any co-medications.

#4. Although the biliary pain was improved after Rowachol administration, 3 cases (#2, #10, and #15 cases in Figure 2) were undergone cholecystectomy. Please explain the reason for these cases required surgical treatment.

Answer: Thank you for pointing it out. The reasons for cholecystectomy have been described in the “Results-Clinical outcome” section (Page 4, line 122-127). Of the 3 cases you mentioned, #2 and #10 cases were referred to “symptoms recurrence after stopping Rowachol treatment (n=2)” and #15 case was referred to “aggravation of GB wall thickening (n=1)”. To help the reader understand, we add the following description to the figure legend 2; “The other three patients finally underwent cholecystectomy due to symptoms recurrence after stopping Rowachol treatment (Case 2 and 10) and aggravation of GB wall thickening (Case 15).” (Page 5, line 140-142)

#5. Please describe the adverse events after Rowachol administration.

Answer: Thank you for your valuable comment. Unfortunately, due to the retrospective nature of the study, it is not possible to accurately assess the adverse events resulting from Rowachol treatment. However, we confirmed that there were no patient who discontinued Rowachol due to its side effects.

#6. Because Rowachol has an anti-bacterial effect, it is necessary to consider the risk of drug-resistant bacteria. Please discuss the duration of administration of Rowachol and the timing of discontinuation.

Answer> Thank you for your meaningful opinion. Although Rowachol was considered to have an anti-bacterial effect in in vitro laboratory experiments, it is not clinically used as an antibiotic, and in my opinion, it would be little concern about the in vivo colonization of drug-resistant bacteria. However, we absolutely agree that it is important to investigate the optimal treatment period for Rowachol administration. In many observational and prospective studies for patients with gallstone disease, including difficult-to-remove CBD stones, and post-cholecystectomy pain, the treatment duration ranged from at least 3 to 24 months. Most studies represented a treatment period of 6 to 12 months. In our personal opinions, Rowachol treatment is usually recommended for more than 6 months, and if patients’ responses were not appropriate after 12 months of treatment, additional examinations or other treatment options such as surgery have been considered. In addition, we added this limitation in “Discussion” section as follows; “Even within the study group, treatment duration and timing of follow-up cholescintigraphy were widely distributed, which can confuse the research results” (Page 7, line 217-218).

#7. Six and 9 cases of Figure 2 were evaluated 2nd GBEF measurement after discontinuation of Rowachol at least 3 months. It may be better to exclude these cases since they are thought to be evaluated after the effect of Rowachol has been lost.

Answer: Thank you for your constructive suggestion. We absolutely agreed that a long interval from discontinue of Rowachol administration to 2nd GBEF measurement could affect the Rowachol’s effect on GBEF. However, we considered it was impossible to determine the optimal interval between discontinue of Rowachol treatment and 2nd GBEF measurement. For example, case 11 in Figure 2 had about 2 months-interval between stop of Rowachol treatment and 2nd cholescintigraphy. This case doesn’t seem much different from case 9. Additionally, there was another innate problem with the difference of time interval between initial and follow-up cholescintigraphy (median 3.5 months; range 2.6-32.6months). Considering these limitations and the purpose of this pilot study, we decided not to exclude any patient according to the timing of follow-up GBEF measurement, regardless of treatment duration. Please understand our intentions with your broad generosity.

Minor

#1. Please explain “Cholescitigrapy” using some figures.

Answer: Thank you for your comment. We inserted an example of representative cholescintigraphy of a patient with improved GBEF as Figure 4 in “Results – Results of cholescintigraphy” section (Page 5, line 153-154 and 161-163).

Round 2

Reviewer 2 Report

I think that the authors well responded to my comments.

Please mention the response to Questions #2 (Is there any case who received endoscopic sphincterotomy after or during Rowachole administration?), and #5 (Please describe the adverse events after Rowachol administration.) in the main text, for the readers' understanding.

Thank you.

Author Response

Please mention the response to Questions #2 (Is there any case who received endoscopic sphincterotomy after or during Rowachole administration?), and #5 (Please describe the adverse events after Rowachol administration.) in the main text, for the readers' understanding.

Thank you.

Answer: Thank you for your advice. We added the phrases and sentences as bellows.

“There was no patient who received the endoscopic sphincterotomy for Sphincter of Oddi disease. Additionally, no one discontinued Rowachol due to the side effects.” (Page 4, line 127~129) (Results – Clinical outcomes section)